# The intelligent evaluation model of the English humanistic landscape in agricultural industrial parks by the SPEAKING model: From the perspective of fish-vegetable symbiosis in new agriculture

Yiping He[1], Mingyue Gao[2]*, Luyao Wang[2]

**1** Basic Teach Office, Shaanxi Fashion Engineering University, Xi'an City, China, **2** Institute of Art and Design, Shaanxi Fashion Engineering University, Xi'an City, China

* 610297670@qq.com

## Abstract

To more accurately capture the expression of the English humanistic landscape in agricultural industrial parks under the emerging agricultural paradigm of fish-vegetable symbiosis, and to address the limitations of unscientific evaluation standards and inadequate adaptability in Chinese-English translation within multimodal contexts, this study proposes an intelligent translation evaluation framework based on the SPEAKING model—comprising Setting, Participants, Ends, Act Sequence, Key, Instrumentalities, Norms, and Genre. The study identifies the core elements essential for articulating the English humanistic landscape of agricultural industrial parks and conducts a comprehensive analysis from the dual perspectives of translation accuracy and adaptability. Fish-vegetable symbiosis, an ecological agricultural system integrating aquaculture and plant cultivation, emphasizes resource recycling and ecological synergy. Internationally referred to as the "aquaponics system," this model has become a pivotal direction in sustainable ecological agriculture due to its efficiency and environmental compatibility. This study investigates multimodal translation tasks across text, image, and speech data. It addresses two primary challenges: (1) the absence of robust theoretical grounding in existing translation evaluation systems, which leads to partial and insufficiently contextualized assessments in agricultural industrial park translations; and (2) difficulties in maintaining consistency and readability across multimodal translation tasks, particularly in speech and visual modalities. The proposed optimization model integrates linguistic theory with deep learning techniques, providing a detailed analysis of contextual translation elements. Comparative evaluations are conducted against five prominent translation models: Multilingual T5 (mT5), Multilingual Bidirectional and Auto-Regressive Transformers (mBART), Delta Language Model (DeltaLM), Many-to-Many Multilingual Translation Model-100 (M2M-100), and Marian Machine Translation (MarianMT). Experimental

**Data availability statement:** All relevant data are within the manuscript and its Supporting Information files.

**Funding:** This work was supported by the General Special Scientific Research Program of Shaanxi Provincial Department of Education in 2023: Research on the Application of the New Aquaponics Agricultural Model in the Taibai Taochuan Agricultural Industrial Park. Project Number: 23JK0306. The funders had no role in study design, data collection and analysis, decision to publish, or preparation of the manuscript.

**Competing interests:** The authors have declared that no competing interests exist.

results indicate that the proposed model outperforms existing benchmarks across multiple evaluation metrics. For translation accuracy, the Setting score for text data reaches 96.72, exceeding mT5's 92.35; the Instrumentalities score for image data is 96.11, outperforming DeltaLM's 93.12; and the Ends score for speech data achieves 94.83, surpassing MarianMT's 91.67. In terms of translation adaptability, the Genre score for text data is 96.41, compared to mT5's 93.21; the Key score for image data is 92.78, slightly higher than mBART's 92.12; and the Norms score for speech data is 91.78, exceeding DeltaLM's 90.23. These findings offer both theoretical insights and practical implications for enhancing multimodal translation evaluation systems and optimizing cross-modal translation tasks. The proposed model significantly contributes to improving the accuracy and adaptability of language expression in the context of agricultural landscapes, advancing research in intelligent translation and natural language processing.

## Introduction

Amid ongoing agricultural modernization and the implementation of the rural revitalization strategy, agricultural industrial parks with distinctive Chinese characteristics are increasingly entering the global stage. As essential platforms for showcasing regional cultural identity and ecological values, the English translation of cultural landscape elements within these parks plays an increasingly vital role in international communication [1,2]. In particular, with the widespread adoption of ecological agricultural models such as aquaponics, the effective and culturally appropriate transmission of associated concepts and connotations has become a critical concern in cross-cultural communication [3]. In practice, Chinese-English translations within agricultural industrial parks often exhibit inconsistencies in expression, semantic imbalance, and limited cultural adaptability. These deficiencies not only hinder the accurate dissemination of information but also weaken efforts to establish a coherent and competitive international brand image for the region. Such issues extend beyond linguistic inaccuracies, reflecting deeper shortcomings in contextual awareness, audience engagement, and communicative intent [4]. At the same time, current translation evaluation mechanisms demonstrate notable limitations [5]. Traditional manual evaluation methods are heavily dependent on subjective judgment, lacking standardization and scalability, and are thus inadequate for addressing the nuanced quality demands of diverse application scenarios. Although automated translation evaluation techniques have continued to evolve, most existing models remain constrained by surface-level semantic similarity metrics. These models generally lack grounding in linguistic theory and are insufficient for assessing the adaptability of translations in complex intercultural contexts. Recent advances in natural language processing (NLP) and machine learning have generated growing interest in intelligent translation evaluation. There is an increasing need for systematic and context-sensitive assessment of translation quality in multimodal and multi-scenario environments. As a result, the integration of linguistic theory into intelligent evaluation frameworks has emerged

as a crucial research direction, with the goal of enhancing both the scientific rigor and contextual relevance of translation assessment.

In response to these challenges, the present study develops an intelligent translation evaluation model for English humanistic landscape texts in agricultural industrial parks, based on the SPEAKING framework—comprising Setting, Participants, Ends, Act Sequence, Key, Instrumentalities, Norms, and Genre. The model is designed to evaluate both the accuracy and cultural adaptability of Chinese-English translations. By employing a systematic analytical approach grounded in linguistic dimensions, the study aims to provide a structured, quantifiable, and context-sensitive method for assessing translation quality. This framework offers both a theoretical basis and a practical tool for improving language communication and enhancing the international image of agricultural industrial parks.

## Literature review

As an important medium for cross-cultural communication, the translation quality and adaptability of English humanistic landscape texts have received increasing attention in academic discourse in recent years. The SPEAKING model, proposed by linguist Dell Hymes, serves as a theoretical framework for analyzing language use within social and cultural contexts. It has been widely applied in sociolinguistic studies and, more recently, in translation research. Pratama et al. (2024) proposed that the seven elements of the SPEAKING model offered an effective means of analyzing cultural and pragmatic features across various communicative contexts, thus providing theoretical support for the study of cultural adaptability in translation [6]. Ding and Sazalli (2024) further argued that incorporating the SPEAKING model into translation evaluation helped identify deficiencies in pragmatic functions and cultural adaptability. However, they noted that a comprehensive indicator framework had yet to be established [7]. Translation accuracy and cultural adaptability are widely recognized as the two core dimensions of translation quality. Recent studies have primarily explored the relationship between linguistic performance and cultural context. Abbasi et al. (2023) emphasized that, in tourism translation, accuracy directly influenced international tourists' comprehension of a destination, while cultural adaptability played a critical role in shaping their cultural experience [8]. Similarly, Crisianita and Mandasari (2022) observed that insufficient cultural adaptability often results in the failure to convey the original text's cultural connotations, particularly in contexts involving frequent intercultural interaction [9]. With rapid advancements in NLP and machine learning, intelligent translation evaluation has become a prominent area of research. Scholars have explored evaluation metrics, algorithmic models, and contextual applications. Jabber and Mahmood (2024) showed that machine learning-based evaluation systems were capable of processing large volumes of translation data efficiently. Nonetheless, they highlighted persisting limitations in evaluating contextual adaptability [10].

In summary, existing research has made notable progress in applying the SPEAKING model, investigating translation accuracy and adaptability, and advancing intelligent translation evaluation models. However, several research gaps remain. The integration of the SPEAKING model with intelligent evaluation technologies has yet to be comprehensively explored. The development of a systematic indicator framework for assessing translation adaptability remains incomplete, and the applicability of intelligent evaluation models across diverse contexts requires further empirical investigation.

## Research model

### Intelligent translation evaluation model framework

To scientifically and comprehensively assess the accuracy and adaptability of English translations of humanistic landscape texts, this study constructs an intelligent translation evaluation model framework based on the SPEAKING model [11,12]. This framework integrates linguistic theory with NLP technologies to enable multidimensional and intelligent assessment of translated content. The SPEAKING model—proposed by Dell Hymes—focuses on communicative context and comprises eight key elements: Setting, Participants, Ends, Act Sequence, Key, Instrumentalities, Norms, and Genre [13,14]. These elements are critically important in translation evaluation and provide a robust theoretical foundation for the

development of relevant evaluation metrics [15–17]. In this study, the SPEAKING model is adapted into specific evaluation dimensions, which are then operationalized and quantified through intelligent technologies. The resulting framework consists of three core modules, as outlined in Table 1.

The model begins by collecting a translation corpus and standardizing the data, followed by the extraction of contextual elements from the text. Intelligent algorithms are then applied to score the translation, generating a comprehensive evaluation that reflects both accuracy and adaptability. The evaluation results are subsequently broken down into specific scores for the seven elements of the SPEAKING model, analyzing the translation's performance across multiple dimensions [21]. Finally, based on the evaluation outcomes, recommendations for improvement are provided, addressing areas where the translation may fall short, and offering practical guidance for translation practice [22]. This intelligent translation evaluation model framework not only holds theoretical significance but also serves as a valuable reference for enhancing the quality of English humanistic landscape translations, thus laying a solid foundation for future research [23,24].

## Intelligent model construction

To achieve a scientifically robust and comprehensive assessment of the accuracy and adaptability of English humanistic landscape translations, this study constructs an intelligent translation evaluation model based on the theoretical framework of the SPEAKING model. By integrating NLP and machine learning technologies, the model demonstrates systematic and innovative characteristics in its theoretical foundation, algorithmic design, and practical application.

The core objective of the intelligent translation evaluation model is to combine linguistic theory with intelligent technology, enabling a multidimensional evaluation of translation texts through quantification and automation [25]. The elements of the SPEAKING model are transformed into quantifiable evaluation metrics, and automated analysis is conducted using intelligent technologies. To ensure the model's adaptability, both the corpus and predefined rules are incorporated, enhancing the model's ability to understand and interpret language context. Designed with a modular structure, the model is adaptable to diverse translation scenarios and language requirements. The architecture of the model is depicted in Fig 1.

The model employs a Transformer-based deep learning architecture to compute semantic similarity and comprehend the contextual nuances of the translation text. Dependency syntax analysis is utilized to identify potential grammatical errors and logical inconsistencies within the translation. Additionally, a multi-class classification model assesses the translation's performance in terms of cultural adaptability. Multi-task learning techniques are employed to simultaneously

**Table 1. Core structure of the intelligent translation evaluation framework.**

| Source | Analysis |
|---|---|
| Data Input Module | • Corpus Collection: Bilingual Chinese-English texts are collected from agricultural industrial park contexts, including signage and promotional materials, to construct a multilingual corpus.<br>• Contextual Information Extraction: Contextual data are extracted using the seven elements of the SPEAKING model, capturing details such as linguistic environment, audience characteristics, and cultural communication objectives [18]. |
| Evaluation Metric Module | • Accuracy Evaluation: This component evaluates the semantic, grammatical, and lexical consistency between the source and target texts.<br>• Adaptability Evaluation: This component assesses the degree of cultural appropriateness, pragmatic alignment, and contextual relevance of the translation. |
| Intelligent Processing Module | • Natural Language Processing Technology: NLP tools and machine learning techniques are applied to conduct automated analysis of translation quality.<br>• Model Algorithms: A hybrid approach combining deep learning algorithms with rule-based methods is used to enable intelligent, scalable evaluation of translation quality [19].<br>• Result Visualization: Evaluation outcomes are displayed through visual charts, offering a clear and interpretable representation of translation performance across different dimensions [20]. |

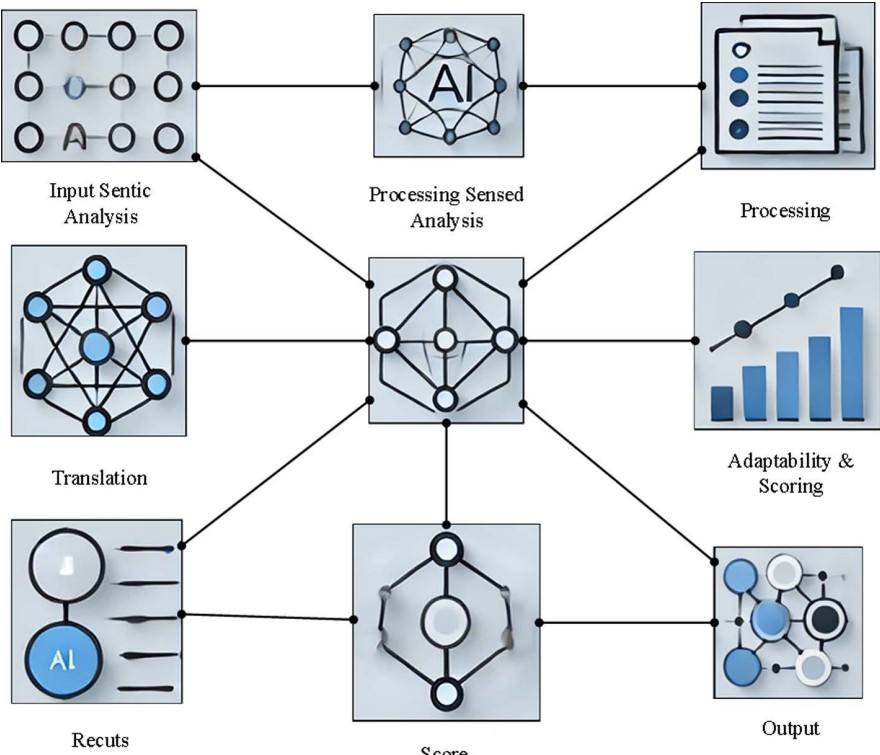

**Fig 1. Intelligent translation evaluation model architecture.**

optimize the evaluation of both translation accuracy and adaptability. A weighted scoring algorithm is applied to quantify various metrics, ensuring a scientifically rigorous and impartial evaluation result.

## Experimental design

The dataset selected for this experiment is the Webster Machine Translation (WMT) Chinese-English Machine Translation Training Corpus. This corpus contains approximately 25 million pairs of Chinese-English parallel sentences, making it suitable for training and evaluating machine translation models. The dataset details are as follows:

(1)  Data Sources: The corpus integrates Chinese-English parallel texts from various sources, including ParaCrawl, News-Commentary, Wiki-Titles, WikiMatrix, among others.

(2) Data Scale: The dataset consists of approximately 25 million sentence pairs in Chinese and English.

(3) Data Format: The dataset is in text format, with source and target languages aligned line by line. Each line corresponds to a pair of translated sentences.

The dataset is available for download from the ModelScope platform (https://www.modelscope.cn/home). For the purposes of this study, the experimental dataset is categorized into three dimensions based on data type: text data, image data, and language data. This classification enables comprehensive coverage of the translation model's diverse application needs, ensuring that both training and evaluation are more precisely aligned with the model's capabilities.

In the sample extraction process, this study carefully considers the contextual differences and practical application characteristics of various data types, ultimately ensuring a balanced and representative sample distribution. The dimensions of the dataset are as follows:

(1) Text Data: This dimension includes 10,000 pairs of Chinese-English sentence pairs, primarily covering common humanistic landscape language found in agricultural industrial parks. This includes promotional slogans, exhibition descriptions, and policy interpretations.

(2) Image Data: This dimension comprises 3,000 pairs of Chinese-English corpora, primarily derived from bilingual-labeled images or scenes. Text is extracted using Optical Character Recognition (OCR) technology and aligned with manually annotated English content.

(3) Speech Data: This dimension includes 3,000 pairs of samples, with data transcribed into text via a speech recognition system and matched with corresponding English translations. Typical contexts covered include guided tours and voice inquiries.

The total sample size amounts to 16,000 pairs, ensuring strong representativeness, comparability, and practicality across different data types.

Below is a portion of the code used in model construction:

```
# Define the dataset class
class TranslationDataset(Dataset):
  def __init__(self, input_texts, labels, tokenizer, max_length):
    self.input_texts=input_texts
    self.labels=labels
    self.tokenizer=tokenizer
    self.max_length=max_length
  def __len__(self):
    return len(self.input_texts)
```

To ensure the stability and reproducibility of the experiment, this study provides a detailed configuration of the hardware:

(1) Processor Model: Intel Xeon Silver 4216

(2) Graphics Card Model: NVIDIA A100

(3) Memory Model: Samsung 128GB DDR4

(4) Storage Device Model: Samsung 970 EVO Plus NVMe SSD

(5) Power Supply Model: Corsair RM850x

For data preprocessing, the vocabulary size is set to 32,000, employing a subword tokenization algorithm to ensure comprehensive coverage of diverse input languages. The maximum sequence length is set to 256 tokens, which accommodates the typical length of image-text and speech transcription data. Padding is applied to the right side of the sequences to maintain consistent alignment. To enhance the model's robustness, data augmentation is performed with a noise ratio of 0.155, applying random replacements, deletions, and other perturbation operations to simulate real-world non-standard expressions. Additionally, a sentence mixing ratio of 0.1 is used to create composite input scenarios across varying contexts, improving the model's adaptability to complex semantics. The base model is built upon the standard Transformer architecture. The hidden layer dimension is 768, with a network depth of 12 layers. Each layer contains 8 multi-head attention mechanisms to capture dependencies between contexts fully. The feedforward layer dimension is 3072, ensuring the model's expressive capacity during non-linear transformations. A dropout rate of 0.1 is applied to mitigate overfitting and improve generalization ability. Additionally, all modules incorporate residual connections and layer normalization to maintain gradient stability throughout training.

The training process is configured as follows:

(1) Learning Rate: Set to 5e-1, dynamically adjusted using a warm-up phase and linear decay strategy to enhance stability during the initial stages of training.

(2) Epochs: The model is trained for 10 epochs to ensure sufficient learning across the various data dimensions.

(3) Batch Size: Set to 64 to balance training efficiency with memory resource utilization.

(4) Gradient Clipping: The gradient clipping threshold is set to 1 to prevent extreme gradients from causing model divergence.

(5) Validation Set: To ensure differential evaluation of training and testing performance, 10% of the training data is set aside as a validation set. This subset is used to monitor model performance after each epoch and guide early stopping decisions.

The models compared in the experiment include Multilingual T5 (mT5), Multilingual Bidirectional and Auto-Regressive Transformers (mBART), Delta Language Model (DeltaLM), Many-to-Many Multilingual Translation Model-100 (M2M-100), and Marian Machine Translation (MarianMT).

## Intelligent translation evaluation model experiment evaluation

### Performance evaluation

This study evaluates the experiment from two dimensions: efficiency and resource usage, and stability and user experience. Each dimension includes four evaluation metrics. Translation speed, measured as the number of tokens a model can translate per unit of time, is a key factor in assessing the model's responsiveness and suitability for real-time scenarios such as smart navigation systems and online translation services. High response capabilities are critical for enhancing user experience and system throughput. Model latency, on the other hand, refers to the average time required for the model to produce a translation result after receiving input. It is particularly useful for evaluating performance in low-latency environments, such as voice interactions. Models with low latency are better suited for deployment in mobile devices, edge computing, or interactive systems. Memory usage refers to the amount of memory required by the model during its operation. It is directly related to the model's adaptability across different hardware environments, such as cloud-based systems, embedded devices, or lightweight equipment. Memory optimization is especially important for devices with limited resources. Data scalability refers to the model's ability to maintain performance when processing large-scale input data, including accuracy variation and resource consumption stability. Models with high scalability can handle complex and diverse translation tasks, offering greater potential for practical deployment and expansion.

The experimental results for the efficiency and resource usage dimension are shown in Fig 2:

The results in Fig 2 indicate that the optimized model performs exceptionally well in translation speed across different data dimensions. In the text data dimension, the optimized model achieves a translation speed of 2675.347 tokens per second, surpassing all other models, including mBART at 2417.564 tokens per second and DeltaLM at 2321.892 tokens per second. In the image data dimension, the optimized model's speed is 2436.215 tokens per second, second only to M2M-100 at 2543.783 tokens per second, and ahead of MarianMT at 2189.342 tokens per second. For the speech data dimension, the optimized model achieves a speed of 2237.891 tokens per second, which is better than mT5 at 2119.234 tokens per second. Regarding model latency, the optimized model shows the lowest latency in the text data dimension at 12.782 ms, which is significantly better than mT5's latency of 18.321 ms. In the image data dimension, the latency of the optimized model is 15.672 ms, which is close to M2M-100 and lower than mBART at 21.892 ms. In the speech data dimension, the optimized model's latency is 20.452 ms, outperforming DeltaLM, which has a latency of 27.134 ms.

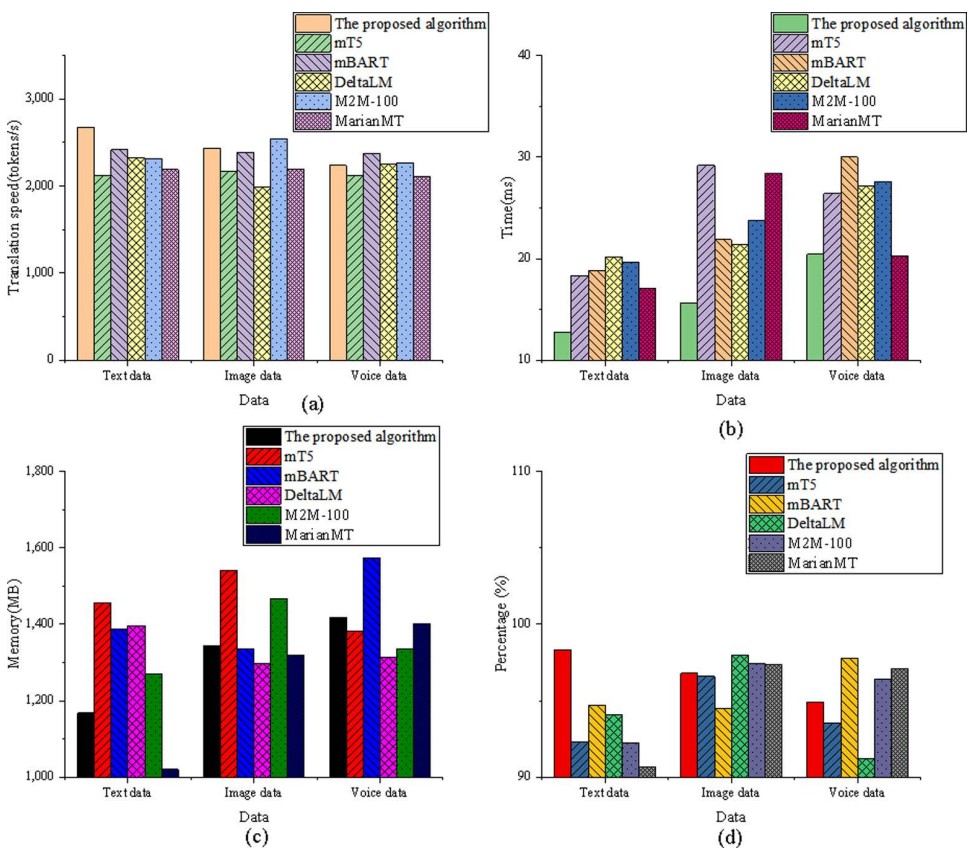

**Fig 2. Evaluation of efficiency and resource usage dimensions (a) translation speed; (b) model latency; (c) memory usage; (d) data scalability.**

In terms of memory usage, the optimized model requires 1167.342 MB in the text data dimension, which is lower than mT5 at 1456.231 MB but higher than MarianMT at 1019.564 MB. For the image data dimension, the memory usage of the optimized model is 1342.789 MB, slightly higher than DeltaLM at 1298.341 MB. In the speech data dimension, the optimized model requires 1418.237 MB, which is better than mBART at 1572.892 MB. Finally, when considering data scalability, the optimized model scores 98.324% in the text data dimension, which is significantly higher than mBART at 94.712%. In the image data dimension, the scalability of the optimized model is 96.785%, just slightly lower than M2M-100 at 97.421%. For the speech data dimension, the optimized model scores 94.892%, outperforming DeltaLM, which scores 91.214%.

The stability and user experience dimensions are essential for assessing a translation model's reliability and user satisfaction under varying conditions. Error recovery ability, translation consistency, language diversity support, and user readability are the key metrics in this evaluation. Error recovery ability reflects the model's resilience and its capacity to produce accurate translations despite input errors, such as typos or missing words. This is particularly valuable for real-world applications where non-standard or noisy inputs are common. Translation consistency measures the uniformity of the model's output when translating the same or similar inputs multiple times, ensuring stable results across different contexts. This is crucial for standardized tasks, such as translating technical terms, brand names, or official documents. Language diversity support assesses the model's ability to handle diverse linguistic structures, cultural contexts, and expression modes, particularly when dealing with the translation of culturally specific content like English translations of cultural landscapes in agricultural parks. The model must adapt to both Western and Eastern expression styles and

naming conventions. Lastly, user readability is a subjective evaluation, based on user ratings or expert assessments, that focuses on the fluency, grammatical correctness, and naturalness of the translations. This metric directly correlates to the end user's acceptance and satisfaction with the model's output. The results for these factors are shown in Fig 3. The comparison results for the stability and user experience dimension are shown in Fig 3:

In the error recovery ability comparison, the optimized model outperforms MarianMT in the text data dimension with a score of 89.214%, significantly higher than MarianMT's 84.341%. For the image data dimension, the optimized model scored 86.892%, closely matching mT5 at 86.721%. In the speech data dimension, the optimized model scored 87.341%, surpassing mBART at 83.782%. In terms of translation consistency, the optimized model scored 94.892% in the text data dimension, outperforming DeltaLM at 91.213%. In the image data dimension, the optimized model scored 93.124%, slightly lower than M2M-100 at 94.231%, while in the speech data dimension, it scored 91.892%, surpassing MarianMT at 89.712%. Regarding language diversity support, the optimized model scored 93.214% in the text data dimension, just under mBART's 94.782%. In the image data dimension, the optimized model scored 92.124%, outperforming DeltaLM's 88.782%, and in the speech data dimension, it scored 91.341%, surpassing MarianMT at 89.231%. Finally, in the user readability comparison, the optimized model received a score of 4.871 in the text data dimension, the highest among all models. In the image data dimension, it scored 4.791, higher than mT5 at 4.678, and in the speech data dimension, it scored 4.681, ahead of M2M-100 at 4.521.

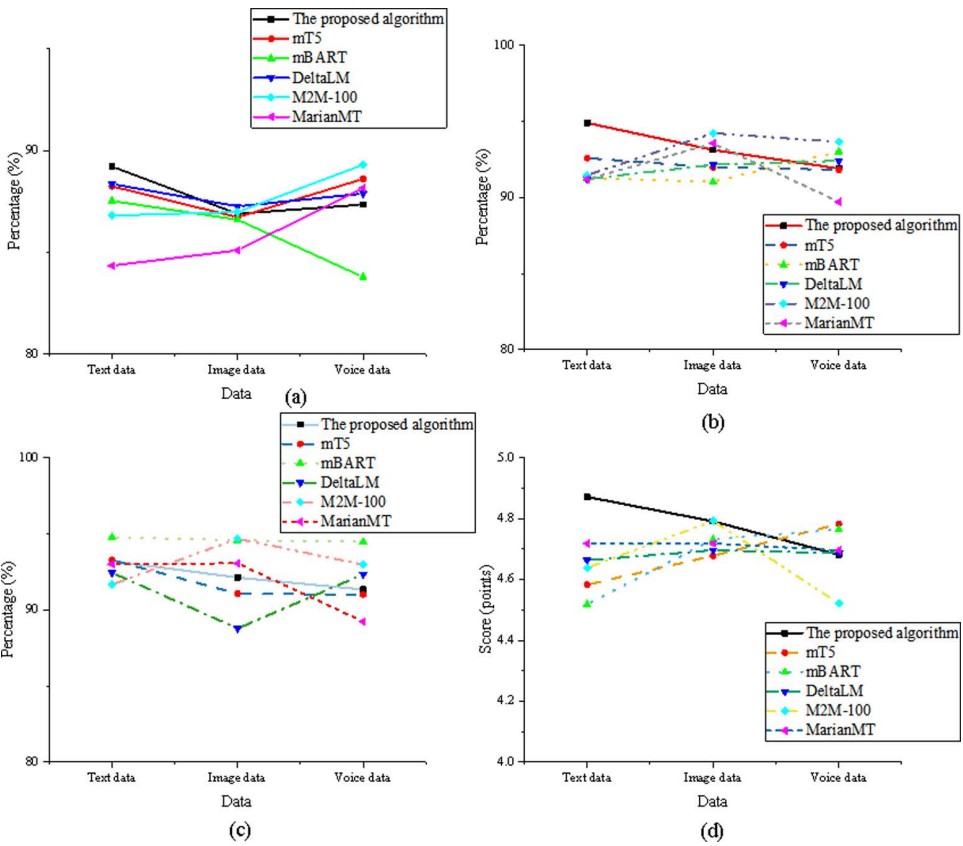

**Fig 3. Comparison of stability and user experience dimensions (a) error recovery ability; (b) translation consistency; (c) language diversity support; (d) user readability.**

## Analysis of the intelligent evaluation results of translation Corpora

The study experiments based on the elements of the SPEAKING model, dividing the eight comparison elements into Setting, Participants, Ends, Act Sequence, Key, Instrumentalities, Norms, and Genre. The comparison results for the Setting dimension are shown in Fig 4:

In Fig 4, regarding translation accuracy, the optimized model in the text data dimension scored 96.72, higher than DeltaLM's 92.35. In the image data dimension, the optimized model scored 94.83, slightly lower than M2M-100's 95.32. In the speech data dimension, the optimized model scored 93.45, higher than MarianMT's 91.23. In the adaptability analysis, the optimized model in the text data dimension scored 94.82 for translation adaptability, significantly higher than mT5's 90.32. In the image data dimension, the optimized model scored 93.71, close to M2M-100's 94.12. In the speech data dimension, the optimized model scored 92.34, higher than DeltaLM's 89.76.

The results in Fig 5 show that, in translation accuracy, the optimized model in the text data dimension scored 97.12, higher than mT5's 93.45. In the image data dimension, the optimized model scored 95.23, similar to mBART. In the speech data dimension, the optimized model scored 94.51, higher than DeltaLM's 92.32. In the adaptability analysis, the optimized model in the text data dimension scored 95.12, higher than mT5's 91.56. In the image data dimension, the

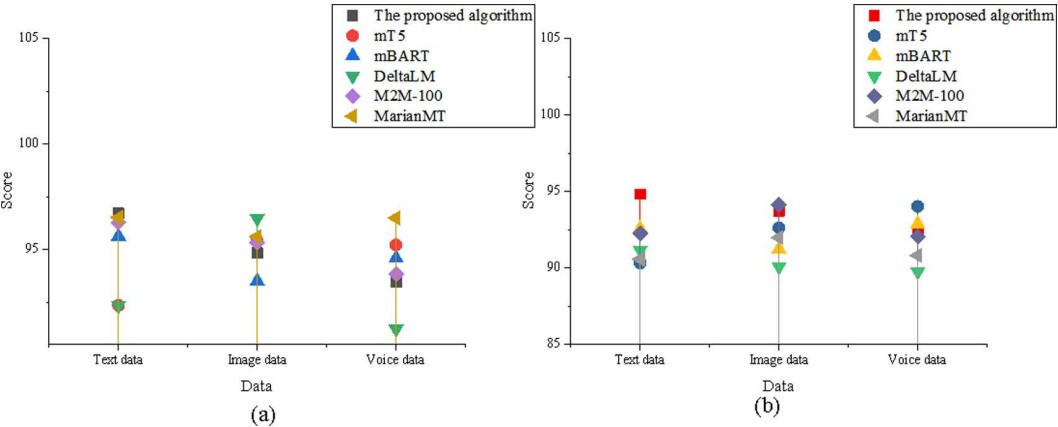

**Fig 4. Comparison of the setting element (a) translation accuracy; (b) adaptability.**

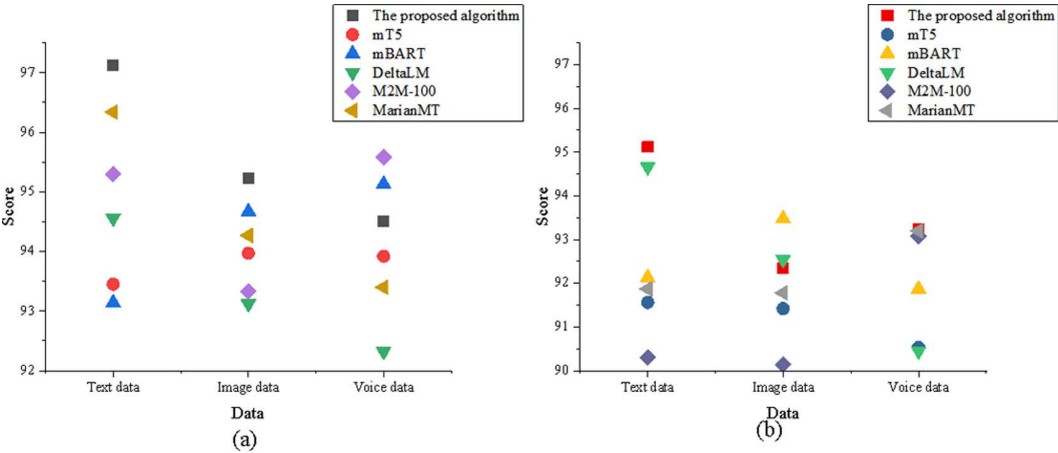

**Fig 5. Comparison of the participants element (a) translation accuracy; (b) adaptability.**

optimized model scored 92.34, higher than MarianMT's 91.78. In the speech data dimension, the optimized model scored 93.23, significantly higher than DeltaLM's 90.45.

In Fig 6, regarding translation accuracy, the optimized model in the text data dimension scored 98.12, making it the best-performing model in this dimension. In the image data dimension, it scored 95.72, outperforming MarianMT's 93.21. In the speech data dimension, it scored 94.83, leading mT5. In terms of adaptability, the optimized model in the text data dimension scored 96.21, the highest among all models. In the image data dimension, it scored 94.11, higher than mBART's 92.78. In the speech data dimension, the optimized model scored 93.41, significantly outperforming MarianMT's 91.67.

In Fig 7, the results show that the optimized model demonstrates strong performance across all data dimensions in terms of both translation accuracy and adaptability. For translation accuracy, the optimized model scores 96.52 in the text data dimension, significantly outperforming mBART, which scored 91.24. In the image data dimension, it achieved a score of 94.32, closely matching the performance of M2M-100. In the speech data dimension, the model scored 93.81, surpassing MarianMT's 90.45. Regarding adaptability, the optimized model attained a score of 95.56 in the text data dimension, again exceeding mBART's 92.34. In the image data dimension, the model achieved 93.67, which is comparable to

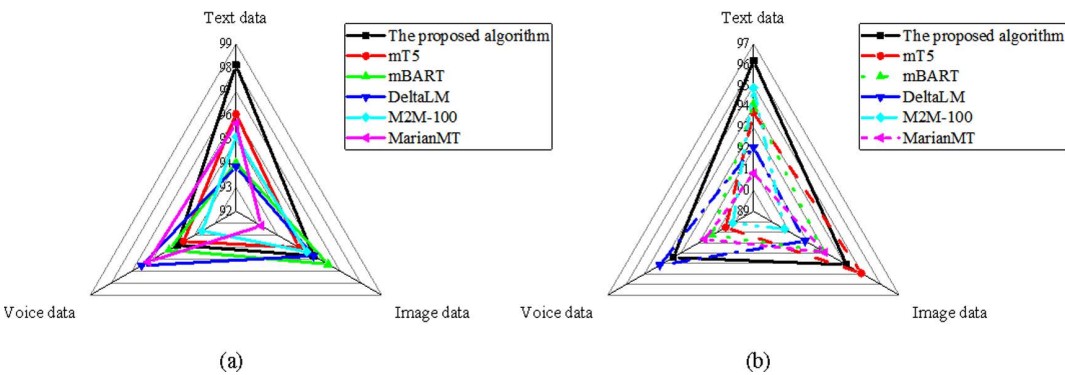

**Fig 6. Comparison of the ends element (a) translation accuracy; (b) adaptability.**

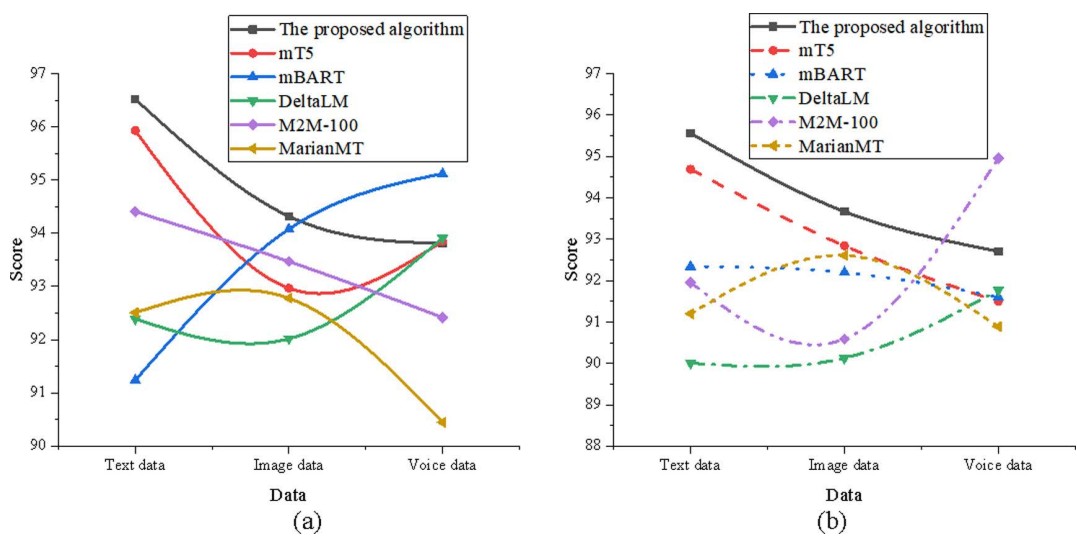

**Fig 7. Comparison of the act sequence element (a) translation accuracy; (b) adaptability.**

M2M-100. For the speech data dimension, the optimized model recorded a score of 92.71, demonstrating superior adaptability compared to DeltaLM's 90.89.

The data in Fig 8 shows that the optimized model maintains strong overall performance across translation accuracy and adaptability metrics in all data dimensions. In terms of translation accuracy, the optimized model scored 97.41 in the text data dimension, significantly outperforming DeltaLM's 92.34. In the image data dimension, it achieved a score of 95.12, which is slightly below M2M-100 but still within a competitive range. For the speech data dimension, the model scored 94.54, clearly surpassing mBART's 91.76. Regarding adaptability, the optimized model achieved 94.91 in the text data dimension, significantly higher than mT5's 92.12. In the image data dimension, the model scored 92.78, which is close to mBART's 92.12, indicating comparable adaptability. In the speech data dimension, the model recorded a score of 91.56, outperforming MarianMT's 90.34, thus demonstrating superior generalization in diverse speech input contexts.

In Fig 9, the optimized model demonstrates leading performance in both translation accuracy and adaptability across all data dimensions. For translation accuracy, it achieved a score of 98.45 in the text data dimension, the highest among all models. In the image data dimension, the model scored 96.11, outperforming MarianMT's 92.73. In the speech data

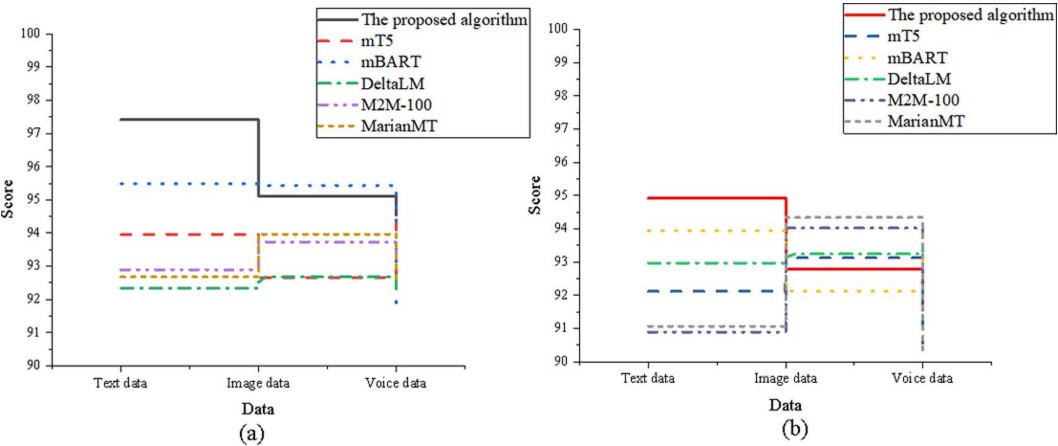

**Fig 8. Comparison of the key element (a) translation accuracy; (b) adaptability.**

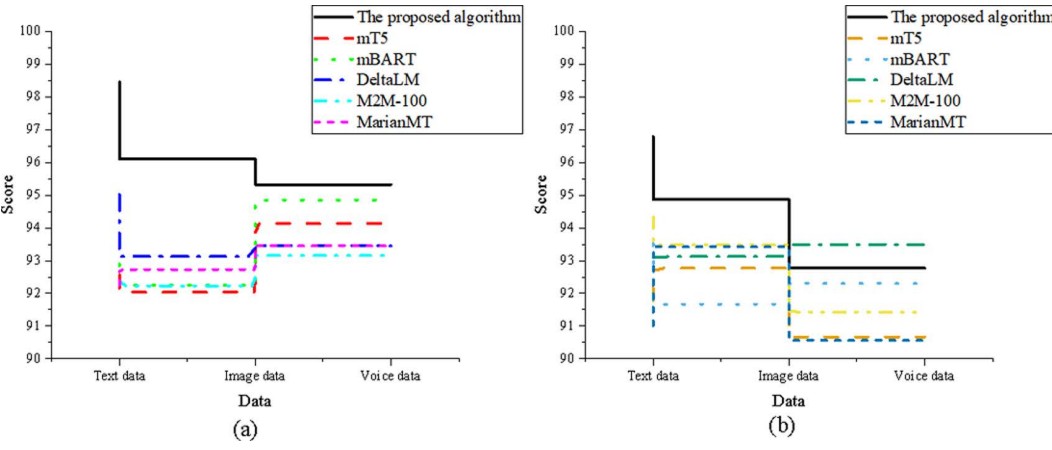

**Fig 9. Comparison of the instrumentalities element (a) translation accuracy; (b) adaptability.**

dimension, it reached 95.32, surpassing DeltaLM's 93.45. Regarding adaptability, the optimized model also ranks highest in the text data dimension with a score of 96.78. In the image data dimension, it achieved 94.87, exceeding DeltaLM's 93.12. In the speech data dimension, it scored 92.78, ahead of M2M-100's 91.34.

In Fig 10, the optimized model continues to exhibit strong performance in both translation accuracy and adaptability across the text, image, and speech data dimensions. For translation accuracy, the model scored 96.82 in the text data dimension, outperforming mT5's 93.21. In the image data dimension, it achieved a score of 94.63, comparable to mBART. In the speech data dimension, the score reached 93.91, exceeding MarianMT's 91.54. Regarding adaptability, the optimized model scored 95.34 in the text data dimension, higher than mT5's 92.45. In the image data dimension, it recorded 93.45, closely aligned with mBART's 92.78. In the speech data dimension, the model achieved 91.78, significantly outperforming DeltaLM's 90.23.

In Fig 11, the optimized model maintains its superior performance in translation accuracy and adaptability across all three data dimensions. For translation accuracy, it scored 97.23 in the text data dimension, outperforming DeltaLM's 92.87. In the image data dimension, it achieves a score of 95.01, comparable to M2M-100. In the speech data dimension, it recorded 94.32, surpassing mBART's 91.45. Regarding adaptability, the optimized model demonstrated strong robustness with a score of 96.41 in the text data dimension, notably higher than mT5's 93.21. It also performed well in the image

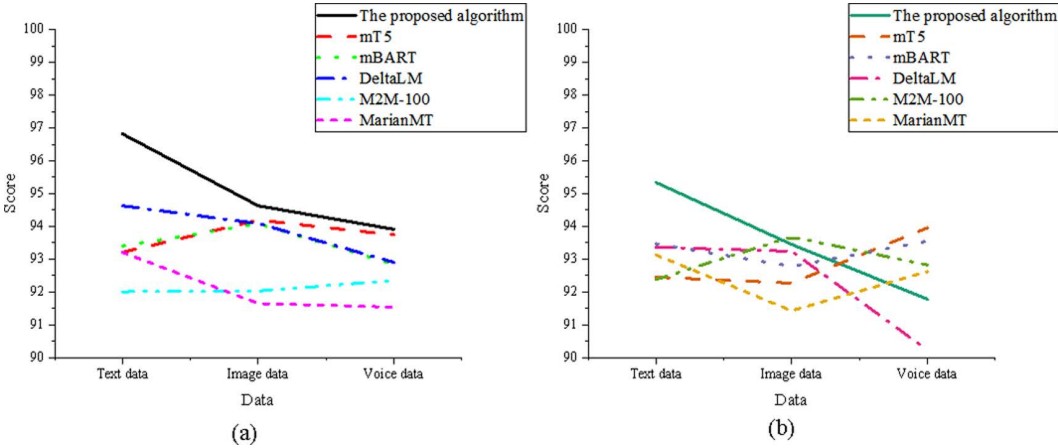

**Fig 10. Comparison of the norms element (a) translation accuracy; (b) adaptability.**

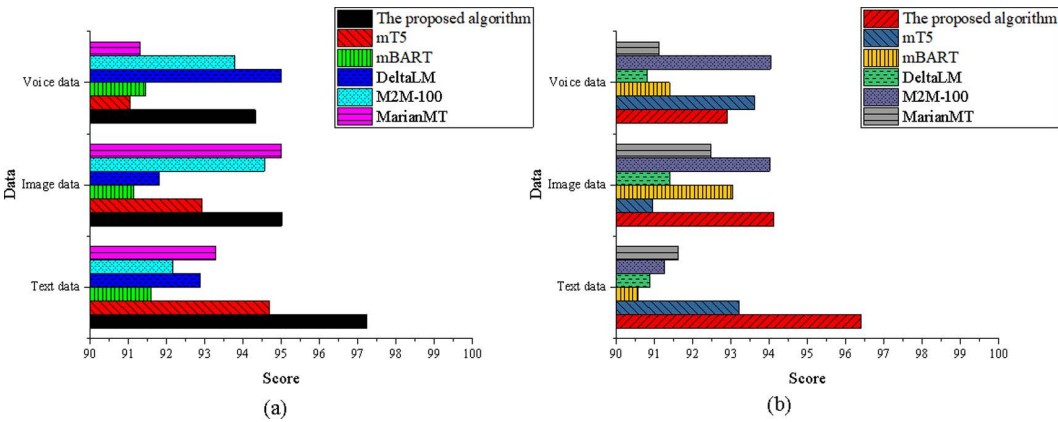

**Fig 11. Comparison of the genre element (a) translation accuracy; (b) adaptability.**

data dimension with a score of 94.12, on par with M2M-100's 94.01, and achieved 92.91 in the speech data dimension, significantly outperforming MarianMT's 91.12.

## Discussion

The experimental results demonstrate that the proposed optimized model consistently delivers stable and superior performance across multiple evaluation dimensions. In terms of efficiency and resource utilization, the model exhibits remarkable translation speed and low response latency. Whether processing text, image, or speech inputs, it ensures rapid responsiveness, underscoring its suitability for real-time multimodal translation tasks. Although its memory usage for image and speech data is slightly higher than that of some lightweight models, it remains within an acceptable range, indicating a well-balanced trade-off between performance and resource consumption. When handling large-scale or non-standard inputs, the model shows strong data scalability and error recovery capabilities, effectively managing complex inputs and confirming its practical value and engineering feasibility. Regarding stability and user experience, the model demonstrates notable strengths in translation consistency and linguistic diversity support. It maintains high translation quality across different languages and contexts, adapts well to diverse linguistic structures and cultural nuances, and produces natural, fluent expressions in cross-linguistic communication. In terms of user readability, the optimized model surpasses baseline models across various input scenarios, indicating that its outputs better align with the linguistic preferences and reading habits of target users, thereby enhancing the overall user experience. Supported by an intelligent evaluation framework, the model's advantage in translation accuracy is particularly pronounced. In the text data dimension, it achieves the highest scores in several fine-grained components, such as Key and Instrumentalities, consistently outperforming models like mT5 and DeltaLM. For image-based data, although certain indicators are slightly lower than those of M2M-100, the model maintains competitive performance, particularly in adapting to context-specific language. In the speech data dimension, it delivers stable outputs and demonstrates robust performance in components such as Instrumentalities and Ends, highlighting its strength in speech-to-text translation. With respect to translation adaptability, the optimized model performs robustly across all evaluated dimensions. In the textual domain, it achieves top scores in nearly all indicators, reflecting strong competence in understanding text structures, pragmatic functions, and cultural contexts. For image-based inputs, its performance is comparable to M2M-100 in components such as Setting and Genre, demonstrating effective adaptability in text-image integration scenarios. In contrast, models such as DeltaLM exhibit noticeable fluctuations across dimensions, revealing limitations in managing complex communicative contexts.

In summary, the optimized model proposed in this study exhibits stable and comprehensive performance advantages across multimodal and multidimensional translation evaluations, reflecting strong generalization capability and significant potential for practical application. Its consistent performance across text, image, and speech modalities indicates suitability for real-world deployment in diverse translation scenarios. Future research may focus on further enhancing the model's adaptability to image and speech inputs, with the aim of extending its applicability and improving its upper-bound performance in increasingly complex and varied translation tasks.

## Conclusion

This study proposes an intelligent translation evaluation framework based on the SPEAKING model, offering a comprehensive approach to assess Chinese-English translation quality in multimodal translation tasks. The framework emphasizes two core dimensions: translation accuracy and translation adaptability. Through experimental comparisons, the optimized model demonstrates exceptional performance across various indicators, particularly excelling in translation accuracy and adaptability evaluations. It exhibits strong stability and adaptability across text, image, and speech data dimensions. The experimental results indicate that the optimized model effectively balances translation accuracy and adaptability in multimodal scenarios, with notable superiority in several refined indicators within the text data dimension. In the translation of image and speech data, the optimized model also showcases high robustness and adaptability,

achieving performance levels comparable to or exceeding those of the best models. By integrating deep learning techniques with linguistic theories, the optimized model provides a nuanced evaluation of complex contextual translation tasks, offering new insights into cross-modal translation evaluation.

However, this study has some limitations. In evaluating translation adaptability for image and speech data, certain indicators score slightly lower than those of comparison models (such as M2M-100), indicating potential for improvement in the model's adaptability to multimodal context modeling. Additionally, this study primarily focuses on Chinese-English translation tasks, without an in-depth examination of translation performance for low-resource languages, which limits the applicability of the optimized model to multilingual tasks. Future research could enhance the optimized model's translation performance in complex multimodal scenarios by incorporating additional modalities (e.g., video, embedded interaction data) and improving multimodal fusion techniques. Furthermore, for translation tasks involving low-resource languages, future studies should explore methods such as transfer learning and cross-lingual pretraining models to improve translation quality and expand the model's applicability.

## Supporting information

**S1 File. Data.**
(ZIP)

## Author contributions

**Conceptualization:** Yiping He, Luyao Wang.

**Data curation:** Yiping He, Luyao Wang.

**Formal analysis:** Yiping He.

**Funding acquisition:** Mingyue Gao.

**Investigation:** Luyao Wang.

**Methodology:** Yiping He, Luyao Wang, Mingyue Gao.

**Project administration:** Yiping He.

**Resources:** Mingyue Gao.

**Software:** Yiping He.

**Supervision:** Yiping He, Mingyue Gao.

**Validation:** Luyao Wang, Mingyue Gao.

**Visualization:** Yiping He, Luyao Wang.

**Writing – original draft:** Yiping He, Luyao Wang, Mingyue Gao.

**Writing – review & editing:** Yiping He, Luyao Wang, Mingyue Gao.

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
