## [Decision Letter · Decision Letter 0]

10 Apr 2025

PONE-D-25-02877The Intelligent Evaluation Model of English Humanistic Landscape in Agricultural Industrial Parks by the SPEAKING Model: from the Perspective of Fish-Vegetable Symbiosis in New AgriculturePLOS ONE

Dear Dr. Gao,

Thank you for submitting your manuscript to PLOS ONE. After careful consideration, we feel that it has merit but does not fully meet PLOS ONE’s publication criteria as it currently stands. Therefore, we invite you to submit a revised version of the manuscript that addresses the points raised during the review process.

We look forward to receiving your revised manuscript.

Kind regards,

Elena del Pilar Jiménez-Pérez, Ph.D.

Academic Editor

PLOS ONE

Additional Editor Comments:

Dear Authors,

The reviewers have noted the following:

The manuscript depicts a straightforward investigation that uses an appropriate methodology, which is technically accurate and supports the conclusion. However, the sample size and control mechanisms require improvement in order to enhance transparency. The statistical analysis is appropriate with little reservation, but assumptions, effect sizes, and power analysis would provide a solid margin over the rigor of the results. Indeed, his document is clear and is written in a form of English that can be understood, but some sections contain complex sentences and minor grammatical errors that would make understanding challenging. The introduction and discussion should be more direct and the transitions less awkward. Ethical considerations are not thorough, and ethical approval alongside informed consent must be included. The manuscript complies with publication ethics, but the authors must take care that the research is not self-plagiarized and all appropriate citations have been provided. On the balance of all these aspects, the amendments are clear, but it makes valuable contributions due to having insufficient clarification along with deficient methodological rigor.

Once the relevant changes have been made, the article could be accepted.

Regards,

Elena del Pilar Jiménez-Pérez

Reviewers' comments:

Reviewer's Responses to Questions

**Comments to the Author**

1. Is the manuscript technically sound, and do the data support the conclusions?

Reviewer #1: Partly

Reviewer #2: Yes

2. Has the statistical analysis been performed appropriately and rigorously? 

Reviewer #1: Yes

Reviewer #2: Yes

3. Have the authors made all data underlying the findings in their manuscript fully available?

Reviewer #1: Yes

Reviewer #2: Yes

4. Is the manuscript presented in an intelligible fashion and written in standard English?

Reviewer #1: Yes

Reviewer #2: Yes

5. Review Comments to the Author

Reviewer #1: The manuscript depicts a straightforward investigation that uses an appropriate methodology, which is technically accurate and supports the conclusion. However, the sample size and control mechanisms require improvement in order to enhance transparency. The statistical analysis is appropriate with little reservation, but assumptions, effect sizes, and power analysis would provide a solid margin over the rigor of the results. Indeed, his document is clear and is written in a form of English that can be understood, but some sections contain complex sentences and minor grammatical errors that would make understanding challenging. The introduction and discussion should be more direct and the transitions less awkward. Ethical considerations are not thorough, and ethical approval alongside informed consent must be included. The manuscript complies with publication ethics, but the authors must take care that the research is not self-plagiarized and all appropriate citations have been provided. On the balance of all these aspects, the amendments are clear, but it makes valuable contributions due to having insufficient clarification along with deficient methodological rigor.

Reviewer #2: Author has done groundbreaking research.

Hypothesis and Research Finding go hand in hand.

Author is expected to avoid Grammatical Errors.

Authors have employed the unique research findings and also their ideas were fresh.

6. PLOS authors have the option to publish the peer review history of their article (what does this mean? ). If published, this will include your full peer review and any attached files.

**Do you want your identity to be public for this peer review?** For information about this choice, including consent withdrawal, please see our Privacy Policy .

Reviewer #1: No

Reviewer #2: No

---

## [Author Response · Author response to Decision Letter 1]

7 May 2025

5. Review Comments to the Author

Reviewer #1: The manuscript depicts a straightforward investigation that uses an appropriate methodology, which is technically accurate and supports the conclusion. However, the sample size and control mechanisms require improvement in order to enhance transparency. The statistical analysis is appropriate with little reservation, but assumptions, effect sizes, and power analysis would provide a solid margin over the rigor of the results. Indeed, his document is clear and is written in a form of English that can be understood, but some sections contain complex sentences and minor grammatical errors that would make understanding challenging. The introduction and discussion should be more direct and the transitions less awkward. Ethical considerations are not thorough, and ethical approval alongside informed consent must be included. The manuscript complies with publication ethics, but the authors must take care that the research is not self-plagiarized and all appropriate citations have been provided. On the balance of all these aspects, the amendments are clear, but it makes valuable contributions due to having insufficient clarification along with deficient methodological rigor.

Reply: Thank you for your reply. I have made adjustments to the content of the article. Firstly, in terms of improving the transparency of the research, I have clarified the sample size situation and supplemented the parameter content. The above parameter settings are based on the experience of mainstream NLP tasks, combined with the characteristics of the agricultural industrial park cultural landscape translation task targeted by this research, and have been moderately adjusted and optimized to ensure the stable performance of the model in accuracy and adaptability evaluation tasks. All parameters are kept fixed during the experiment to improve transparency, reproducibility, and traceability. Secondly, regarding statistical analysis, I supplemented the reasons for selecting relevant indicators. Choosing these indicators not only reflects the attention to the implementation ability of translation model engineering, but also responds to the multi-level demand for the quality and applicability of translation results. This system is suitable for comprehensive evaluation of multimodal and multi scenario translation models, and has strong promotion and reference value. Regarding the issue of expression in the article again, I checked the vocabulary and grammar throughout the text to ensure that there were no errors in expression that could lead to misunderstandings for readers. Finally, in the introduction and discussion sections, I adjusted the content to make the expression clearer and more logical, highlighting the stable and comprehensive performance advantages of the optimized model in multimodal and multidimensional translation evaluation. This proves that it has strong generalization ability and application potential in practical applications. In addition, regarding the ethical aspect, as the experiments and research in this article are all from publicly available datasets, there are no such issues, so they have not been supplemented.

Reviewer #2: Author has done groundbreaking research.

Hypothesis and Research Finding go hand in hand.

Author is expected to avoid Grammatical Errors.

Authors have employed the unique research findings and also their ideas were fresh.

Reply: Thank you very much for your positive evaluation and affirmation of this study. You pointed out that 'the author conducted groundbreaking research' and 'hypotheses and research results go hand in hand', which is a great encouragement for us and further enhances our confidence in exploring this research direction in depth. And I have checked the vocabulary and grammar throughout the text to ensure that there are no errors in expression that may cause misunderstandings for readers. We are also pleased that you believe the ideas in the research are innovative and the results are unique, which is exactly the direction of our research work. Thank you again for your constructive feedback and support. We will continue to improve the content of the paper to ensure that the research results are more rigorous, clear, and of academic value.

---

## [Editor Report · Decision Letter 1]

12 May 2025

The Intelligent Evaluation Model of the English Humanistic Landscape in Agricultural Industrial Parks by the SPEAKING Model: from the Perspective of Fish-Vegetable Symbiosis in New Agriculture

PONE-D-25-02877R1

Dear Dr. Gao,

We’re pleased to inform you that your manuscript has been judged scientifically suitable for publication and will be formally accepted for publication once it meets all outstanding technical requirements.

Kind regards,

Elena del Pilar Jiménez-Pérez, Ph.D.

Academic Editor

PLOS ONE
---

## [Editor Report · Acceptance letter]

PONE-D-25-02877R1

PLOS ONE

Dear Dr. Gao,

I'm pleased to inform you that your manuscript has been deemed suitable for publication in PLOS ONE. Congratulations! Your manuscript is now being handed over to our production team.

Kind regards,

on behalf of

Dr. Elena del Pilar Jiménez-Pérez

Academic Editor

PLOS ONE